# Can a Community-Led Intervention Offering Social Support and Health Education Improve Maternal Health? A Repeated Measures Evaluation of the PACT Project Run in a Socially Deprived London Borough

**DOI:** 10.3390/ijerph17082795

**Published:** 2020-04-18

**Authors:** June Brown, Ana Luderowski, Josephine Namusisi-Riley, Imogen Moore-Shelley, Matthew Bolton, Derek Bolton

**Affiliations:** 1Institute of Psychiatry, Psychology and Neuroscience, King’s College London, London SE5 8AF, UK; ana.luderowski@kcl.ac.uk (A.L.); derek.bolton@kcl.ac.uk (D.B.); 2Citizens UK, 112 Cavell Street, London E1 2JA, UK; josephine.namusisiriley@citizensuk.org (J.N.-R.); imogen@connectionsinmind.co.uk (I.M.-S.); matthew.bolton@citizensuk.org (M.B.)

**Keywords:** maternal health, community engagement, mental health, depression, anxiety, social support, health literacy, PACT, Citizens UK

## Abstract

Social adversity can significantly influence the wellbeing of mothers and their children. Maternal health may be improved through strengthened support networks and better health literacy. Health improvement at the population level requires optimizing of the collaboration between statutory health services, civic organizations (e.g., churches, schools), as well as community groups and parents. Two key elements in improving community engagement are co-production and community control. This study evaluated a co-produced and community-led project, PACT (Parents and Communities Together), for mothers in a deprived south London borough. The project offered social support and health education. Intended effects were improvements in mental health, health literacy, and social support, assessed by standardized measures in a pre-post design. Sixty-one mothers consented to take part in the evaluation. Significant improvements were found in mental health measures, in health literacy, for those with low literacy at baseline, and in overall and some specific aspects of social support. Satisfaction with the project was high. We found that the project engaged local populations that access statutory health services relatively less. We conclude that community-organized and community-led interventions in collaboration with statutory health services can increase accessibility and can improve mothers’ mental health and other health-related outcomes.

## 1. Introduction

There is increasing evidence that social adversity has detectable effects on children, with these found early in development as well as later in childhood [1,2,3]. The importance of caring for the health of mothers, not only for the sake of mothers but also for their children, is widely recognized [2,4,5,6]. Ways of improving maternal health include strengthening their support networks [2,7,8], and improving health literacy [9,10]. In order to reduce the onset and burden of disease in populations facing high levels of social adversity, more effort and funding has to be devoted to prevention by improving the social determinants of health, particularly for young children [1,2,3,4,5,6,11]. Social determinants of health refer to social conditions, including early childhood development, access to good-quality education, and decent living conditions. These approaches need to be applied at the community level, prior to referral to specialist health services. The importance of communities being involved in their own health is widely recognized. It is likely that population-level health improvement requires transformation of the health service delivery by optimizing the use of community social capital. This would involve developing co-operative social networks among statutory services (e.g., midwifery and health visiting services, children’s centers), civic organizations (e.g., churches, mosques, schools), as well as community groups and parents themselves. The UK National Institute for Health and Clinical Excellence in its public health guidance makes specific recommendations for enhancing community engagement, key themes within which are co-production and community control [12].

This paper describes the evaluation of a project developed for mothers, which was community organized and community led, involving mothers, statutory health services (e.g., midwives), and local civic organizations (e.g., churches). The project was based in the south London borough of Southwark, which has high levels of multiple deprivations. The present project follows a pilot study that demonstrated the feasibility of local community leadership in constructing a social support project with local mothers, in the course of which the participating mothers requested health education workshops. The pilot study showed promising results in improving maternal mental health and some aspects of social capital [13]. In the subsequent larger project reported here, a more structured health education program as well as social support system has been offered. The community groups involved changed the name of the project, from “Strengthening Babies’ Futures” as it was in the pilot project, to Parents and Communities Together (PACT). There were two main objectives of the PACT project: (1) To use community-organized and -led methods to engage women from mothers from local populations that tend to access statutory health services relatively less, and (2) to improve maternal mental health and other health-related outcomes.

The UK policy context of the PACT project, including public health objectives and recommended community engagement strategies, and the background health science, is referred to above [1,2,3,4,11,12]. While many of the public health objectives and community engagement strategies are common internationally, there are variations between countries and localities in objectives, strategies, and implementation. Our purpose here is not to review these variations but to focus on the PACT project as a case study implementing public health policy in a particular locality. Consistent with this, the PACT project was not primarily a research project but was rather the provision of a new service for local mothers, which, if it were to bring about the anticipated benefits, and given available community health funding, might be extended and implemented elsewhere. The purpose of the evaluation, the research study, was to assess whether and to what extent the project achieved the anticipated benefits. Both the PACT project and the evaluation study are reported here.

The results of the evaluation study were that the main objectives of the PACT project were broadly achieved. We conclude that community organizing is an effective way of engaging communities in health projects, including members of populations that tend to access statutory services relatively less, and that engaged communities can work well with statutory health services to co-produce and deliver interventions, such as social support and health education, to benefit health.

## 2. Materials and Methods

### 2.1. Collaborative Partners in the PACT Project and Aim of the Project

Like the pilot study, PACT involved collaboration between Citizens UK [14], the largest community organizing charity in the UK, and King’s Health Partners, an Academic Health Sciences Centre [15]. Citizens UK uses the ‘broad-based’ community organizing model and methodology, which is well theorized, deriving from the work of Saul Alinsky in Chicago [16,17] and which is applied by community organizations throughout the US. Key features of the approach include building trust-based reciprocal relationships among individuals in already existing communities, particularly civic institutions (e.g., churches, schools), fostering networks among diverse institutions (health, voluntary/statutory, academic), developing community leadership, and working towards goals decided by communities. The broad-based community organizing approach is well-suited to optimize community engagement using the principles of co-production and community leadership as recommended by the UK National Institute for Health and Clinical Excellence as noted above. As in the pilot study, research clinical psychologists in King’s Health Partners provided advice on evidence-based methods of reducing psychosocial stress, contributed to developing health education workshops, and led on the evaluation methodology. New to this larger phase was the involvement of community maternity services, midwifery, and health visiting, bringing with them their health knowledge and skills to the collaboration and evaluation.

The broad aim of this collaboration between Citizens UK and King’s Health Partners is to make use of the resources, leadership, social capital, and potential peer leadership in existing civic institutions (e.g., schools, churches) and to combine this with community health services (e.g., midwifery, health visiting) and clinical academic resources to translate evidenced-based health improvement technologies to benefit local mothers and children. It also uses the tools of community organizing to build the capacity of local parents and communities to work with local statutory partners to provide social support for parents and to improve parental mental health and other health-related outcomes and thereby improve children’s developmental outcomes.

### 2.2. Community Involvement and Evolving Co-production

#### 2.2.1. From the Beginning to the Pilot Project

Citizens UK’s member organizations decide in local meetings their common purposes and priorities, and which ones are to be adopted as projects. Around 2009, health appeared among local priorities in South London Citizens. Various community health issues were raised in these early stages, and the upshot was a decision to focus on supporting new mothers and mothers with young children. These deliberations and common purposes were discussed with clinical academic collaborators at the (then) Institute of Psychiatry in Southwark and maternity services in King’s Health Partners. There was a shared appreciation that a condition of such a project affecting policy and being adopted by health commissioners would be robust evaluation of a primary health outcome. There was a shared appreciation that expertise on community engagement lay with Citizens UK, that health research methodology expertise lay with clinical academics and clinicians, each, however, consulting with the other, while the choice of key outcomes of interest, what the project was trying to achieve, development of relevant evidenced-based interventions, content, and content delivery had to involve all stakeholders in co-production. This division and cooperation of work and expertise ran through all the stages of the project as it evolved.

It was integral to the rationale of the project that community assets would be brought to bear, in the provision of accommodation, advice, support, and volunteer time. However, funding was required for a part-time Citizens UK professional organizer plus evaluation costs. South London has a major health charity, the Guy’s and St Thomas’ Charity, an independent urban health foundation with the primary remit of improving the health of the people of Lambeth and Southwark, and whose strategic goals at that time included community engagement to support community health and reduce health inequalities. We were advised to apply for funding for a pilot project to demonstrate the feasibility of engaging communities in a health project and evaluation. In the early stages, senior staff of London Citizens had meetings with leaders of one member institution in South London, a local Baptist church, to discuss and agree the broad aims of the then proposed project. The prior determined intervention was only that interested members of the church and part-time London Citizens professional community organizers would work together to facilitate forming a local new mothers’ group, with the aim of increasing social support. A proposal, including an evaluation plan, was accepted in 2013 and ran for a year.

#### 2.2.2. The Pilot Study

The process of engagement into the project and providing social support from that point was started by the church community and two part-time professional community organizers (total 0.8 whole time equivalent). Key features of the process that emerged from that starting point included: A team of about five or six volunteer local community leaders and the two community organizers helped make links with other local institutions, drawing partly on previous existing relationships; the participating institutions grew rapidly to include three churches, one Islamic Centre, one faith-based charity that ran a large mother and toddler group, one after-school project, and one youth club; the expanding team of local community leaders and the two part-time community organizers also worked together to seek out mothers, pregnant or with children under 2 years, and invited them to take part; they approached mothers in participating organizations and made contact with other mothers by door knocking on local estates and in public spaces, such as parks. They also gave public talks and then followed up with interested individuals. Some mothers were signposted to the program by a friend or community leader/teacher/professional. Approximately 25 women were asked and wanted to participate to varying extents. What emerged once the mothers began to meet was that they formed their own social support network. They planned regular meetings, initially spaced by a week or two, then weekly. In advance of their meetings, the participating mothers agreed on relevant topics to discuss, such as breast feeding, sleep routines, relationships with partners, managing stress, housing, and juggling work and child care. The women began to make requests to the organizations involved, which were positively responded to, such as requests to the participating civic institutions for rooms and facilities for meeting, and to health providers for educational classes on parenting, diet, and child development, as well as information talks from early years providers. In summary, the community-led ‘intervention’ that evolved comprised mothers meeting together to provide mutual social support, choosing discussion topics, concerns and worries, and sharing advice, these sessions being supplemented by requested health information and educational workshops. In addition, there were six meetings on the on-going evaluation of the project, including on average five participating mothers who wanted to be involved in its implementation. Further details of the pilot project are reported elsewhere [13].

#### 2.2.3. Co-Production Phase of the Main Study

On the basis of the promising results of the pilot study [13], the same funder was willing to consider an application for the main study. To facilitate this, the funder also approved an application for funding for a 6-month co-production phase to draw together the various stakeholders to plan the larger project together and write the application. These included the group of mothers who participated in the pilot project, continuing to meet and by now calling the group “Mumspace”, becoming key leaders by this stage, local community leaders, senior local maternity services staff, and the clinical academics. Further crucial work in the co-production phase was consultation with local National Health Service (NHS) commissioning and NHS national policy agencies. The resulting funding application for the main study was accepted. The co-production phase of the main study and the funding application specified the setting, the principles and aims of engagement and participation, and set the broad parameters of the intervention—social support groups and health education workshops—as well as determining the evaluation methodology. The delivery of subsequent engagement and arrangements for the social support groups were community led, and health education content was co-produced and co-delivered as described below.

### 2.3. Setting and Engagement

The project was set in an inner-city London borough, focusing on two electoral wards with high levels of social deprivation and immigration. The intervention was based in three local hubs: One church, one church-related center, and one community center. The PACT staff team comprised four paid part-time staff: A community organizer who was also the project manager (0.6 whole time equivalent, wte), a health visitor (0.6 wte), and two group leaders (0.6 wte and 0.4 wte) who were mothers from the area. In addition, the project was staffed by five volunteers at a hub at any one session to help with childcare. In addition, there were local PACT Parent Champions, parents/key individuals from participating civic organizations (e.g., a children’s center, a local primary school, churches and mosques) and community organizations (e.g., charities). They were trained to take on a new role in their local community, speaking to other parents and signposting them to PACT and other local services in the borough, promoting the benefits of using such services for their children and the wider family. 

The PACT project was open to any mother living in the area who wished to attend. A total of 425 mothers participated over the 30 months of the project. Mothers self-referred after being sign-posted, often by one of the Parent Champions, from nearby civic organizations (e.g., churches, mosques), schools, and statutory health provisions (e.g., local health visiting teams, midwives, maternity clinics, Children’s Centers), ‘leafleting’ or word of mouth or personal invitation. As part of the project there was also a “Baby Bank” at one of the sites, where child clothing, equipment, and accessories were donated and available to mothers in need. Mothers who initially came for the donated baby supplies were invited to PACT by volunteers, group leaders, and other participating mothers.

### 2.4. Components of the Project—The “Intervention”

The main components of the PACT intervention were social support, provided through meetings the mothers chose to call “Mumspace”, and health education. Health education events were co-designed with parents, health visitors, and midwives; some were co-led by parents and health professionals and some by parents only.

The social support groups (Mumspace) were held weekly and ran for 2 h for the duration of the project. There were ongoing weekly groups at each of the three local hubs. Parents spent part of the session playing with their children and talking to other parents over a cup of tea and then part of the time in a separate space when volunteers helped with their children. They also participated in parent-led workshops, on such topics as parenting, immunizations, importance of play and going back to work, motherhood, caring for children, parenting, and personal concerns. Topics for workshops were decided by participants in quarterly meetings. The greater joint decision-making created a welcoming and non-judgmental feel that reduced barriers that some communities report when accessing local statutory provision. Key characteristics of the social support component of the intervention were: Regular frequent meetings (weekly) of sufficient duration (around two hours) in a large space (a community hall), with volunteers on hand to look after children part of the time (around one hour), so that participating mothers have time both to be together and talk together casually while playing with the children, and to be together as mothers without the children to talk over matters of interest to them, about parenting and their own lives, structured into workshops on topics that are chosen by the mothers and led by them.

Heath education events were also provided in the “Parent University”. This comprised a 12-week health education course in weekly sessions of 2 h, coordinated by a health visitor and co-designed with parents, health visitors, and midwives, and was co-led by parents and professionals. Parents with older children co-facilitated the group, with an emphasis on peer sharing rather than didactic teaching. Each session comprised a talk on a topic followed by discussion. Topics covered included birth and childcare, mental health aspects in parenting, how hormones impact on feelings, health behaviors that benefit the baby’s healthy development, nutrition, infant learning, parenting skills, and minor ailments. Mothers graduated at the end of the course if they had completed 8 out of the 12 sessions. Key characteristics of the program of health education workshops were the co-production of topics and content and co-delivery by the mothers and a professional healthcare worker with the relevant expertise (such as a health visitor, who in the UK are qualified nurses/midwives with additional training in community public health nursing, providing services for individuals, families, groups, and communities, including for all children 0–5 years); other professional expertise (such as child psychologists and employment specialists) was brought in as required.

### 2.5. Evaluation

We evaluated the PACT project using two strategies, with different sampling frames and designs. In the evaluation reported here, the sampling frame was mothers who accessed PACT as described above, by local community-organized and -led contacts, and the research design was assessment before and after the intervention. The design was a repeated measures design with two time-points, pre- and post-intervention. For the second evaluation study, we used a quasi-experimental case control design, with the sampling frame of mothers attending midwifery services who accessed PACT following recruitment into a research project. This second evaluation study raised important methodological questions and will be reported separately [18].

For the pre-/post evaluation study reported here, participants were recruited from the 425 mothers participating in the project. In total, 61 participants were recruited for the evaluation, on the basis of the sample size calculations given below. Inclusion criteria were that participants had to be attending the PACT project, though for less 2 months, having accessed the project by local community contacts, to be over 18, female, the parent of at least one child, and with sufficient English to complete the questionnaires. This language inclusion criterion, to emphasize, refers to selection for the evaluation not participation in the PACT project itself.

For the sample size calculations, we used pilot study data on the effect size found for the GHQ-12 (General Health Questionnaire, 12 item) [13]. These showed a pre- to post-difference of 4.9 units (standard deviation (SD) = 9.7), equating to a standardized mean difference of d = 0.51. Based on this effect size found in the pilot study, the required sample size for the study was 34 (significance level 5% and power 80%). The attrition was zero in the pilot study, but we anticipated 10% attrition, increasing the required total sample target size to 38. However, in order to achieve similar group size to that being used in the second case-controlled evaluation, we increased the number recruited to this evaluation to approximately 60.

Mothers eligible for inclusion in the evaluation, engaging consecutively in the PACT project, were invited to take part in the evaluation from May 2016 and recruitment stopped when the target sample size was reached (at 61) in June 2017.

### 2.6. Measures

#### 2.6.1. Sociodemographic Data

A main aim of the evaluation was to investigate the characteristics of mothers who were engaged in the PACT project by local communities, specifically whether community-organized and -led methods would engage mothers from local populations that access statutory health services relatively less. Such populations are sometimes referred to by the expression ‘difficult (or hard) to engage (or reach)’, defined in various ways [19,20], and sometimes with the risk of negative connotation [21]. The local council had commissioned two research projects that identified ‘hidden populations’ within the borough in terms of low response to the census, found to correlate with low engagement with local services, related to poor language skills, shifting households, lack of awareness of services, and immigration concerns [22,23]. We therefore aimed to assess the extent to which women from these locally identified ‘hidden populations’ engaged with the PACT project. For this purpose, we collected self-reported information on participants’ age, place of birth, ethnicity, occupation, relationship status, number of children, first language, and partner’s occupation status. Occupational class was computed by analyzing the employment status and occupation title of the household member with the highest classification, using official national UK guidelines [24].

#### 2.6.2. Engagement with PACT Project

Engagement with the intervention was measured using participants’ attendance to the sessions offered by PACT. This included Mumspace, Parent University, and any other PACT workshops or events. The time period measured was from the participant’s first session with PACT to the follow-up meeting 6 months later. Levels of engagement were defined as: 1–2 sessions attended = ‘not engaged’, 3–4 sessions attended = ‘somewhat engaged’, and 5 or more sessions attended = ‘fully engaged’.

#### 2.6.3. Maternal Mental Health

We used two measures of common mental health problems with good reliability and validity: The Generalized Anxiety Disorder Questionnaire (GAD-7) and the Patient Health Questionnaire (PHQ-9). The GAD-7 is a widely used 7-item measure, which assesses generalized anxiety disorder (=25). A 3-point Likert scale is used: Not at all (0), Several Days (1), More than Half the Days (2), and Almost all the time (3). Scores of 5, 10, and 15 are the cut-offs for mild, moderate, and severe levels of anxiety [25]. The GAD-7 ‘caseness’ threshold is a score ≥8. The Patient Health Questionnaire (PHQ-9) is a widely used 9-item measure, which assesses major depressive disorder [26]. A 3-point Likert scale is used: Not at all (0), Several Days (1), More than Half the Days (2), and Almost all the time (3). Scores of 5, 10, and 15 are the cut-offs for mild, moderate, and severe levels of depression [26]. The PHQ-9 ‘caseness’ threshold is a score ≥10.

In the event that an individual’s rating on these scales suggested need for clinical assessment, the procedure was to raise this issue with the person and plan referral to their general practitioner.

#### 2.6.4. Health Literacy

We used the Newest Vital Sign UK (NVS-UK) [27], a commonly used brief measure with good reliability and validity for assessing health literacy [28]. Six questions assess participants’ ability to interpret health information from a nutritional label on the back of an imaginary carton of ice cream; an incorrect answer is scored 0 and a correct answer is scored 1. The scores are categorized in levels of health literacy: ‘Adequate/high’ (4 or more), ‘limited/intermediate’ (2–3), or ‘low’ (0–1) [22].

#### 2.6.5. Social Capital/Social Support

Social capital is a complex construct comprising various components, including social support [29,30], and there is continued debate about its exact definition and measurement [31,32]. We used several measures for the PACT evaluation, which will be more fully presented in a separate paper. We report here the findings from the Arizona Social Support Interview Schedule (ASSIS), which has good reliability and validity [33,34]. The Arizona Social Support Interview Schedule captures data about many facets of social support networks and their members, and aims to assess perceived social support and quality of support. The seven areas of support are: Intimate interaction, childcare, material support, advice and information, positive feedback, tangible assistance, and socializing. The version administered in this study collected data about the number of network members for each area of support, satisfaction for each area, and demographic information (e.g., age, sex, ethnicity, years known to participant, proximity) about network members. The outcomes analyzed included the total network members for each area of social support, total network size, and total network satisfaction at baseline and follow-up.

#### 2.6.6. Acceptability and Satisfaction with Service

We assessed acceptability and satisfaction using the Social Support Programme Acceptability Rating Scale, previously used in the pilot study of this intervention [13], an adaptation of the Treatment Acceptability Rating Scale, a measure with good reliability and validity [35]. Areas covered include satisfaction with the program offered as well as opportunities to help plan the program. Participants are asked seven questions about different areas of their satisfaction of the program on a scale from 0 (not at all) to 3 (a great deal), with scores summed to a total.

### 2.7. Procedure

PACT staff (described in Section 2.3) initially approached mothers eligible for inclusion in the evaluation about the possibility of being involved in a research project. If the mother agreed to be contacted, the staff member would personally introduce them to one of the researchers. The researcher would explain the project, which involved answering questions at baseline and after 6 months. They then explained the consent process and gave mothers an information sheet and answered any questions about being involved. It was made clear that research involvement was not related to attendance at Mumspace or Parent University, and that they could drop out of the research at any time with no consequences and continue attending. After a period of 24 h, the researcher called the potential participant, and if the mother was still willing to participate, the researcher and participant agreed to meet to sign the consent form and complete the baseline assessment. Assessments usually took place at participants’ houses, private rooms at intervention sites, or private areas in public spaces, such as local cafes or the library. As a compensation for the participants’ time at both baseline and 6-month assessments, £30 shopping vouchers were given. In the 6 months between baseline and follow-up, research participant mothers were treated no differently to non-participants, and attendance was not incentivized or expected. After 6 months, participants, regardless of attendance at PACT, were contacted by the research worker to arrange a follow-up and the baseline assessments re-administered together with the Social Support Programme Acceptability Rating Scale.

Ethical approval for the evaluation was given by the King’s College London Research Ethics Committee, REC Reference number HR15/162334.

### 2.8. Data Analysis Plan

To examine changes between baseline and follow-up scores on the mental health measures, the NVS-UK and the ASSIS were planned to conduct a paired sample t-test. Given an alpha at 0.05, when a calculated t-value is larger than the critical t-value, after considering degrees of freedom (df) for dependent samples (*n* – 1), the null hypothesis will be rejected.

## 3. Results

### 3.1. Participants in the Evaluation

Of the 90 women who were approached to participate in the evaluation, 29 refused and 61 (68%) agreed to take part. Of these 61 participating mothers, 58 (95%) were re-assessed after 6 months. Two women not followed up had moved away by the time of follow-up, and a third could not be re-assessed. Most of the participating mothers lived locally, in Camberwell (52%) and Walworth (23%), wards in the borough of Southwark.

### 3.2. Engagement

Of the 61 mothers participating in the evaluation, 93% engaged with the project to varying extents. Of the 61 participants, 72% were ‘fully engaged’, attending 5 or more sessions during 6 months; 21% were ‘somewhat engaged’, attending 3–4 sessions; and 6.6% did not engage with the project, attending only 1 or 2 sessions.

### 3.3. Sociodemographic Characteristics

The average age of participants was 34 years (SD = 6.1 years), with an age range of 22–53 years. Most (62%) did not have English as their first language. Other self-declared sociodemographic details of mothers in the PACT evaluation study are shown in Table 1.

It can be seen in Table 1 that the majority (51.7%) of participating mothers identified themselves as ethnically black African; approximately equally as from Nigeria and Eritrea. The other main ethnic groups were white British (11.5%), white other (11.5%) who were mainly from Europe, and Latin American (9.8%) who were mainly from Ecuador. The majority of participants were unemployed (62.3%), but when taking into account the participant and partner’s employment status, this was reduced to 37.9% of households being unemployed. The overall demographic pattern of the participant group includes some features consistent with the characteristics of locally identified ‘hidden populations’, understood, as outlined above (Section 2.5), in terms of low response to the census, found to correlate with low engagement with local services, related to poor language skills, shifting households, lack of awareness of services, and immigration concerns.

### 3.4. Maternal Mental Health

Table 2 shows GAD-7-assessed anxiety scores for the whole group and for sub-groups above the ‘caseness’ threshold (≥8) and below the threshold (<8) at baseline and 6-month follow-up, with paired t-test significant differences.

It can be seen in Table 2 that for the 61 participating mothers, the mean baseline GAD-7 score was 6.87 (SD = 5.6), in the mild anxiety range. Approximately one-third scored above the GAD-7 ‘caseness’ threshold. At the follow-up of 58 participants, there was an overall decline in GAD-7 scores that was statistically significant on a paired samples t-test and which equates to a small/medium Cohen effect size of 0.37 [36]. Of the participants whose scores indicated ‘caseness’ at baseline, there was a larger decline in scores, equating to a large Cohen effect of size of 1.67. Among this group, 80% (16/20) recovered to below the GAD-7 ‘caseness’ threshold.

Table 3 shows the PHQ-9-assessed depression scores for the whole group and for sub-groups above the ‘caseness’ threshold (≥10) and below the threshold (<10) at baseline and 6-month follow-up, with paired t-test significant differences.

It can be seen in Table 3 that for the 61 participating mothers, the mean baseline PHQ-9 score was 7.66 (SD = 6.37), in the mild depression range. Just over one-third of the sample scored above the the PHQ-9 ‘caseness’ threshold. At the follow-up of 58 participants, there was an overall decline in the PHQ-9 mean scores that was significant on a paired samples t-test and which equates to a medium Cohen effect size of 0.44. Of the participants whose scores indicated ‘caseness’ at baseline, there was a larger decline, equating to a large Cohen effect of size of 1.65. Among this group, just over 68% (15/22) recovered to below the ‘caseness’ threshold on the PHQ-9.

### 3.5. Health Literacy

Table 4 shows the baseline and follow-up mean scores on the NVS-UK for the whole group (*n* = 55) and sub-groups categorized by baseline scores into low (0–1) (*n* = 13), intermediate (2–3) (*n* = 23), and adequate/high literacy (>3) (*n* = 19), with paired t-test significant differences.

It can be seen in Table 4 that only the sub-group of mothers that had low health literacy at baseline according to the NVS-UK categorization showed significant improvement at follow-up.

### 3.6. Social Support/Social Capital

Table 5 shows the results from the Arizona Social Support Interview Schedule (ASSIS): The mean number of network members for each of the seven areas of support, and total network satisfaction, at baseline and follow-up, with paired t-test significant differences.

It can be seen in Table 5 that there were statistically significant positive changes in the network size for advice/information, intimate interaction, and for pregnancy/childcare support, and for total satisfaction. Reported changes in support with material aid, positive feedback, tangible assistance, socializing, and total support network size were not statistically significant.

### 3.7. Acceptability and Satisfaction with the PACT Project

Table 6 shows the mean scores at follow-up on the six items and total score of the Social Support Programme Acceptability Rating Scale, on a 4-points Likert scale.

It can be seen in Table 6 that high rates of satisfaction were reported on the provision of what was planned, liking the provision, liking co-participants, finding that on balance the project made life better for the person, and in overall satisfaction. There were relatively lower ratings for items relating to involvement with planning the project and making changes to it, perhaps reflecting the fact that these participants joined PACT after the piloting and detailed co-production planning stages.

## 4. Discussion

The PACT community health project involved collaboration between Citizens UK, a community organizing charity, a university, and statutory maternal health services. The broad aim of this collaboration is to encourage community involvement with health and to translate evidenced-based health improvement technologies to benefit local mothers and children. The PACT project was run in Southwark, a borough in London with high levels of social disadvantage. Its specific objectives were, firstly, to use community-organizing methods of community leadership and co-production to engage women from local populations that tend to access statutory health services relatively less, and secondly, to improve maternal mental health and other health-related outcomes. Broadly, the results of this evaluation study suggest that these were broadly achieved, confirming the results of the previous pilot [13]. They suggest that PACT succeeded in engaging a diverse population of mothers, as well as improving health outcomes.

A main aim of PACT was to use community-organizing methods to engage mothers from local populations that access statutory health services relatively less. For the evaluation, we drew on local government research on local “hidden populations”, understood, as outlined above (Section 2.5), in terms of low response to the census, and found to correlate with low engagement with local services, related to poor language skills, shifting households, lack of awareness of services, and immigration concerns. A large number of barriers can hinder groups accessing and engaging with services, including cultural norms and referral obstacles, as well as factors with the service itself [37,38]. The results of the evaluation suggest that the PACT project successfully engaged a diverse group of people from different ethnic groups and that some were over-represented compared with local population base rates. Approximately 50% of the 61 mothers identified themselves as black African, approximately 25% of the 61 from Nigeria, and approximately 10% Latin American. This contrasts with the 2011 Census records, which report only 4.7% Nigerians and 2.7% Latin Americans living in the borough of Southwark [39]. Further, PACT participants have a different self-reported ethnicity profile compared with people attending the local NHS psychology service (Southwark IAPT (Improving Access to Psychological Therapies)), where approximately 60% identified themselves as “white British” [40]. In this NHS service, patients are largely referred by general practitioners (GPs, primary healthcare services) whereas PACT participants are more similar to those who refer themselves, who do not go through their GPs [41], and who are more likely to be more representative of the local population [42].

Several factors may contribute to the PACT project being able to engage with groups that access statutory services relatively less, using community-organizing methods. Community organizing is based in established civic institutions that are already cohesive communities. Citizens UK member institutions collaborating in the PACT project included local mosques and churches with diverse congregations, and which provide supportive services to recent immigrant groups, often unaware of statutory services or who find them hard to reach [22,23]. Secondly, community organizing promotes community engagement by community leadership, collaboration, and grassroots structure. PACT was primarily run by women from the area, involving only one health professional, a health visitor. This promoted increased access, consistent with a commonly found pattern of people preferring to seek informal help from friends, family, and trusted non-professionals rather than professionals [43,44]. Participating mothers also co-produced interventions by choosing, designing, and often running the daily programs. A further consideration is that how mothers perceive their problems also affects help-seeking and engagement. For example, some research suggests that African women are in general are more likely to see mental health problems in terms of social problems rather than as medical [45]. This may be a factor facilitating African women engaging in PACT, because it was set up not as a mental health service but as a community service for all mothers. Importantly, the community-led PACT intervention may provide a feasible gateway to health and social services for populations who may feel more insecure and are less likely to engage with the statutory services.

The results of the evaluation also suggest high engagement with women with anxiety and depression. About a third of the mothers scored above the clinical threshold on the measures used, the GAD-7 and PHQ-9. Data from general population samples is scarce. For a nationally representative German sample, Kocalvent et al. [46] found a prevalence of 5.6% for moderate to high depression, using the PHQ-9 ≥ 10 caseness threshold. NHS data for Southwark 2015–2016 show the prevalence of depression in the adult population as 7.5% [47]. These figures suggest that the levels of depression in participants in the PACT project were relatively high compared with general and local populations.

Participation in the PACT project was associated with significant improvements at 6 months in anxiety and depression in the sample as a whole, equating to a small to medium effect size for anxiety and a medium effect size for depression, with large effects for both conditions for participants whose scores indicated ‘caseness’ at baseline. These effects are comparable with average effect sizes for cognitive behavioral therapy (CBT) for anxiety and depression ascertained in meta-analytic studies [48,49] These results suggest that participation in the project is effective in improving the maternal mental health. There were also significant improvements for participating mothers in some aspects of social support: In the network size for intimate interaction, for pregnancy/childcare support, and for advice/information, as well as total network satisfaction. These results suggest that mothers felt more able to talk to others about personal matters, especially to do with their children, and felt more able to generally get advice and information. This aspect may be helpful in the mental health effects of the intervention being maintained, given that social support and informal help-seeking is a key component of the prevention of depressive problems [50,51].

Regarding health literacy, there were no pre-post improvements as measured by the NVS-UK in the group as a whole. This may be due to most mothers having reasonable levels of NVS-UK-assessed health literacy at the start. However, sub-group analysis showed that the mothers with low literacy improved significantly, and much more than the mothers with high or adequate literacy. The qualitative component of this study, which is separately reported [52], suggests that mothers reported increased confidence to handle their babies following attendance at PACT.

Mothers generally felt satisfied with the program delivered and who delivered it, as assessed by the Social Support Programme Acceptability Rating Scale. There were relatively lower ratings for items relating to involvement with planning the project and making changes to it, perhaps reflecting the fact that these participants joined PACT after the piloting and detailed co-production planning stages.

Limitations of the evaluation include that it used a pre-post measurement design only, and the effects of the passage of time and extraneous factors affecting improvement cannot be excluded. It was with this limitation in mind that we also evaluated PACT using a quasi-experimental case-control design, which required a researcher-led as opposed to a community-led definition of the sampling frame and sampling method. The results of the case-control study showed a very different pattern of findings compared with the evaluation reported here, raising questions about the appropriate design and strategy for evaluating community health projects that are aimed at maximizing engagement, and community organized and led [18]. Notwithstanding the limitations of the evaluation reported here, the results were positive, and the PACT project has been adopted in other parts of London and the UK.

## 5. Conclusions

Community organizing is an effective way of engaging communities in health projects, including members of populations that tend to access statutory services relatively less. Engaged communities can work well with statutory health services to co-produce and deliver interventions, such as social support and health education, to benefit health.

## Figures and Tables

**Table 1 ijerph-17-02795-t001:** Self-declared sociodemographic details of mothers in the Parents and Communities Together (PACT) evaluation study (*n* = 61).

Factors	Categories	% of Sample (*n* = 61)
Self-declared ethnicity	Black African	55.7%
	White British	11.5%
	White any other background	11.5%
	Latin American	9.8%
	Asian	4.9%
Employment status	Unemployed	62.3%
	Working part-time	18%
	Working full-time	16.4%
	Student	3.3%
Relationship Status	Married/living with someone	50.8%
	Single	32.8%
	In a steady relationship	8.2%
	Divorced/separated	8.2%
Household Occupational Class	Professional Managerial and Technical Skilled non-manual Partly skilled Unskilled Unemployed Student	11.5% 11.5% 1.6% 23% 9.8% 36.1% 1.6%
Highest Educational Qualification	Postgraduate degree	9.8%
	Undergraduate Degree	41.0%
	BTEC/NVQ or equivalent	16.4%
	A level or equivalent	11.5%
	GCSE or equivalent	9.8%

**Table 2 ijerph-17-02795-t002:** GAD-7-assessed anxiety scores for whole group and for sub-groups above the ‘caseness’ threshold (≥8) and below the threshold (<8) at baseline and 6-month follow-up with paired t-test significant differences.

Assessments		Number	Mean (SD)	Paired t-test Significant Differences
Overall group				
	Baseline	61	6.87 (5.62)	*p* = 0.001 (t = 3.36, df = 57)
	Follow-up	58	4.76 (3.85)	
Sub-groups above/below ‘caseness’ threshold				
Above threshold	Baseline	21	13.43 (3.87)	*p* < 0.001 (t = 6.57, df = 19)
	Follow-up	20	6.75 (4.66)	
Below threshold	Baseline	40	3.43 (2.37)	ns
	Follow-up	38	3.71 (2.91)	

SD—standard deviation; df—degree of freedom; ns—no significant difference.

**Table 3 ijerph-17-02795-t003:** PHQ-9-assessed depression scores for the whole group and for sub-groups above the ‘caseness’ threshold (≥10) and below the threshold (<10) at baseline and 6-month follow-up with paired t-test significant differences.

Scores		Number	Mean (SD)	Paired t-test Significant Differences
Overall group				
	Baseline scores	61	7.66 (6.37)	*p* < 0.001 (t = 3.78, df = 57)
	Follow-up scores	58	4.83 (4.15)	
Sub-groups above/below ‘caseness’ threshold				
Cases	Baseline scores	23	14.60 (4.44)	*p* < 0.001, (t = 6.17, df = 21)
	Follow-up scores	22	7.23 (4.84)	
Non-cases	Baseline scores	38	3.45 (2.44)	ns
	Follow-up scores	36	3.36 (2.86)	

**Table 4 ijerph-17-02795-t004:** Newest Vital Sign UK (NVS-UK) baseline and follow-up mean scores for the whole group (*n* = 55) and sub-groups categorized by baseline scores into low (0–1) (*n* = 13), intermediate (2–3) (*n* = 23), and adequate/high literacy (>3) (*n* = 19), with paired t-test significant differences.

Add Heading	Baseline Mean (SD)	Follow-up Mean (SD)	Paired t-test Significant Differences
Whole group	2.91 (2.00)	3.02 (2.02)	ns
Low literacy sub-group	0.38 (0.51)	1.54 (1.05)	*p* = 0.003, t = −3.64, df = 13
Intermediate sub-group	2.39 (0.50)	2.48 (1.78)	ns
Adequate/high literacy sub-group	5.26 (0.87)	4.68 (1.67)	ns

**Table 5 ijerph-17-02795-t005:** Arizona Social Support Interview Schedule (ASSIS): The number of network members for each of the seven areas of support and total network size, and total network satisfaction, at baseline and follow-up, and results of the paired t-test, *n* = 58.

Add Heading	Baseline Mean (SD)	6 Months Follow-up Mean (SD)	Paired t-test Significant Differences
No. network members for Intimate Interaction	2.81 (1.83)	3.34 (1.79)	*p* = 0.019 (t = −2.41, df = 57)
No. network members for Pregnancy/Childcare support	1.62 (1.3)	1.98 (2.0)	*p* = 0.049 (t = −2.01, df = 57)
No. network members for Material Aid	1.81 (1.68)	1.91 (2.17)	ns
No. network members for Advice/Information	1.98 (1.66)	3.21 (3.1)	*p* = 0.001 (t = −3.53, df = 57)
No. network members for Positive Feedback	3.6 (5.32)	3.33 (2.77)	ns
No. network members for Tangible Assistance	2.03 (2.32)	2.52 (2.87)	ns
No. network members for Socializing	3.83 (5.42)	4.07 (3.17)	ns
Total network size	7.72 (6.04)	8.28 (4.62)	ns
Total network satisfaction	6.16 (0.82)	6.35 (0.53)	*p* = 0.04 (t = −2.06, df = 57)

**Table 6 ijerph-17-02795-t006:** Mean scores on the Social Support Programme Acceptability Rating Scale: Six items and total score, 4 points Likert scale (0 = not at all; 1= a little: 2 = quite a lot; 3 = a great deal), *n* = 58.

Add Heading	Mean (SD)
Did you feel involved in helping to plan what social support you would find helpful?	1.55 (0.93)
Did you feel able to make changes to the plan to suit your needs during the programme?	1.25 (0.89)
Was the planned social support actually provided?	2.32 (0.72)
Did you like the way the programme was provided to you?	2.54 (0.54)
Did you like the members of the community who were providing the support?	2.73 (0.52)
On balance, did you find that the programme made life better for you?	2.3 (0.81)
In an overall, general sense, how satisfied are you with the programme?	2.59 (0.57)
Total score	15.23 (3.19)

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
