# Peer review of "Can a Community-Led Intervention Offering Social Support and Health Education Improve Maternal Health? A Repeated Measures Evaluation of the PACT Project Run in a Socially Deprived London Borough"

_ijerph, 2020, doi:10.3390/ijerph17082795_

Round 1

Reviewer 1 Report

Manuscript ID: ijerph-736676 Type of manuscript: Article Title: Can a community-led intervention offering social support and health education improve maternal health? a repeated measures evaluation of the PACT project run in a socially deprived London borough

This paper reports the evaluation of a social support and maternal health education project in London. It covers important topics such as maternal mental health, health literacy and community involvement which are critical for informing the evidence-base for effective interventions. However, the paper requires significant editing to support a coherent and sound scientific study. Careful review of grammar throughout is required to help with the flow of the article. 

A succinct approach is required to provide an articulate account of the study details. The wording in some sentences/paragraphs is clunky and needs to be re-written clearly and directly. For example: Line 445 “The upshot, reported elsewhere, was a very different pattern of results….” This doesn’t make sense as a standalone sentence. 

The title suggests the study is a repeated measures evaluation design, yet the study design is not referred to again in the article. Please provide rationale for this design and explanation of methods to match design – if indeed this is a repeated measures evaluation.  

Can you provide a description of the intervention in a diagram/framework to assist with clear understanding of the key characteristics of the intervention? If the intervention were to be replicated – what are the elements that are part of the fidelity of the intervention? It is not clearly described. 

The introduction only cites relatively old references (numbers 1-7). It requires updating to provide a sound understanding of the current evidence. The discussion would also benefit from more recent references including peer-reviewed papers.  

Reference to previous pilot work does not allow for the reader to gain a comprehensive understanding of the community-involvement. In Table 6, women did not rate involvement/planning highly, therefore can the authors claim community involvement at all stages?

Line 168 Why were mothers who require support with English excluded? Surely they would comprise a large component of this community who would benefit from the intervention?  

Line 170. This is the first time the GHQ-12 is mentioned. Need some context around this. 

Line 242 Who were the PACT staff? Were any of them from the community and therefore supporting a community-led approach? 

Line 285 Table 1. Wouldn’t it be more helpful to write maternal country of birth rather than ethnicity so that the demographics are much comparable to other international studies? Did you collect number and age of children?

Line 318 Was referral to specialist care/support provided for mothers with depressive symptoms? 

Was there any consultation conducted regarding choice of measures and their use? This would be an important component of any community-involved research project and especially given 62% did not have English as a first language. Given the NVS-UK health literacy findings, do the authors consider it was an appropriate measure to use? Discussions about the selected measures and their performance would be warranted, particularly in the context of a diverse population. 

Author Response

Cover letter explaining revisions

We thank Reviewer 1 for their helpful comments and suggestions.  Responses are below, with numbering added for convenience. Reference to line numbers are to the numbering of the Revised text.

Derek Bolton, on behalf of all co-authors

COMMENT/SUGGESTION #1: This paper reports the evaluation of a social support and maternal health education project in London. It covers important topics such as maternal mental health, health literacy and community involvement which are critical for informing the evidence-base for effective interventions. However, the paper requires significant editing to support a coherent and sound scientific study. Careful review of grammar throughout is required to help with the flow of the article. /  A succinct approach is required to provide an articulate account of the study details. The wording in some sentences/paragraphs is clunky and needs to be re-written clearly and directly. For example: Line 445 “The upshot, reported elsewhere, was a very different pattern of results….” This doesn’t make sense as a standalone sentence. 

AUTHORS’ RESPONSE: Line 445 has been corrected (lines 550-51) and other sentences improved in the revisions.

COMMENT/SUGGESTION #2: The title suggests the study is a repeated measures evaluation design, yet the study design is not referred to again in the article. Please provide rationale for this design and explanation of methods to match design – if indeed this is a repeated measures evaluation.  

AUTHORS’ RESPONSE: We have added an explicit statement of the study design in the first paragraph of the evaluation section, lines 253-257.

Assessments pre-/post-intervention are standardly used to assess effects of interventions. The main subsidiary question is typically whether to have a control group within this kind of design, and this is covered in the same paragraph (we did that as well, reported elsewhere).  Regarding terminology, statements that repeated measures designs and analyses can use only two time-points are commonly to be found. E.g. at

https://www.statisticssolutions.com/matching-your-participants-to-ensure-a-successful-pre-post-test/

“Pre-post test designs, also known as repeated measures designs, involve the repeated measurement of the same individuals at two (or more) timepoints. Repeated measures designs allow for a statistically powerful analysis of changes in a measure over time, or to assess the effect of an intervention.”

COMMENT/SUGGESTION #3: Can you provide a description of the intervention in a diagram/framework to assist with clear understanding of the key characteristics of the intervention? If the intervention were to be replicated – what are the elements that are part of the fidelity of the intervention? It is not clearly described. 

AUTHORS’ RESPONSE: This is a very helpful suggestion for addition thank you. We  are unsure how to represent the intervention in a diagram, but have added sentences specifying key characteristics of the intervention, lines 226-232 and 242-47.

COMMENT/SUGGESTION #4: The introduction only cites relatively old references (numbers 1-7). It requires updating to provide a sound understanding of the current evidence. The discussion would also benefit from more recent references including peer-reviewed papers.  

AUTHORS’ RESPONSE: We have added more recent references in the introduction, lines 37-43 and 69-71.  The great majority of the papers cited in the Discussion are peer reviewed, the exceptions being government population surveys, and we have added other recent  references, lines 378-81, 521-23 & 534.

COMMENT/SUGGESTION #5: Reference to previous pilot work does not allow for the reader to gain a comprehensive understanding of the community-involvement. In Table 6, women did not rate involvement/planning highly, therefore can the authors claim community involvement at all stages?

AUTHORS’ RESPONSE: We are grateful to the Reviewer for this comment. We have added a new subsection that sets out community involvement at the different stages of developing the project, lines 113-89.  The new subsection explains the piloting and co-production stage and this now provides the context for and makes sense of the existing comment in the text on Table 6 (“There were relatively lower ratings for items relating to involvement with planning the project and making changes to it, perhaps reflecting the fact that these participants joined PACT after the piloting and detailed co-production planning stages.”).

COMMENT/SUGGESTION #6: Line 168 Why were mothers who require support with English excluded? Surely they would comprise a large component of this community who would benefit from the intervention?  

AUTHORS’ RESPONSE: The English language criterion refers to inclusion in the evaluation (N-61), not to inclusion in/access to the intervention (eventual N = 425). We have rewritten the text and included material to make this clearer, lines 259-265.

COMMENT/SUGGESTION #7: Line 170. This is the first time the GHQ-12 is mentioned. Need some context around this. 

AUTHORS’ RESPONSE: We have re-ordered the sentences to make clearer that this was the measure used in the pilot project, lines 266-267.

COMMENT/SUGGESTION #8: Line 242 Who were the PACT staff? Were any of them from the community and therefore supporting a community-led approach? 

AUTHORS’ RESPONSE: A list and description of the PACT staff – 4 paid part-timers, two of whom were members of the local community, and many volunteers, all of whom were local mothers, is given in lines 194-201. We have made it more explicit that these were the staff.  There is a further reference to ‘PACT staff’, line 344, and we have now added a reference back to where this is explained.

COMMENT/SUGGESTION #8: Line 285 Table 1. Wouldn’t it be more helpful to write maternal country of birth rather than ethnicity so that the demographics are much comparable to other international studies? Did you collect number and age of children?

AUTHORS’ RESPONSE: we used self-declared ethnicity, choices including such as “Black African” and “White British” consistent with practice in the UK NHS, but also international practice in research on public health and inequalities, e.g. by the EU (https://ec.europa.eu/newsroom/just/item-detail.cfm?item_id=112035) and in the US (e.g. Mays et al. 2003. Classification of Race and Ethnicity: Implications for Public Health. Annual review of public health. 24. 83-110. 10.1146/annurev.publhealth.24.100901.140927). We did collect data on number (and ages) of children, but there were complexities that made brief description difficult (e.g. some children living for short or long periods with relatives) and we have not reported them.

COMMENT/SUGGESTION #9: Line 318 Was referral to specialist care/support provided for mothers with depressive symptoms? 

AUTHORS’ RESPONSE: Yes and this is now noted, lines 313-14.

COMMENT/SUGGESTION #10: Was there any consultation conducted regarding choice of measures and their use? This would be an important component of any community-involved research project and especially given 62% did not have English as a first language. Given the NVS-UK health literacy findings, do the authors consider it was an appropriate measure to use? Discussions about the selected measures and their performance would be warranted, particularly in the context of a diverse population. 

AUTHORS’ RESPONSE: WE have included in the added material on co-production, in response to Comment/Suggestion #5, that while the key outcomes of interest and the interventions and were co-produced, the research evaluation methodology, including choice of measures for the outcomes of interest, was designed by the clinical academics and health professionals in the project team, lines 124-29. Regarding English-speaking and use of measures, we ensured that participants in the evaluation had sufficient English for this purpose, but this being a condition only of participation in the evaluation, not taking part in PACT, as now made clearer in response to Comment/Suggestion #6. We are not sure how to discuss the performance of the selected measures; the study was not designed to investigate this, e.g. by using multiple measures of the same construct. We have not added this point to the revised text, but could do so if requested.  About the NVS-UK in particular, we do not know the reason why failed to detect change, except for the baseline low literacy group, and we have noted that that this may be due to most mothers having reasonable levels of NVS-UK assessed health literacy at the start (a ‘ceiling effect’), lines 535-37.  

Reviewer 2 Report

Overall the paper is well written, with project settings, interventions, measures and results clearly presented. However, there are still a few areas to be improved.

  • In the introduction part, more evidence on the role of health education and social support in improving maternal health is recommended, as it provides a theoretical basis for this project design, as well as a review of the existing evidence on this paper hypothesis. 
  • Data analysis description not enough, please more detail, for example,  if t-test is a good chioce when mean more than SD ( table 4 "Low literacy sub-group"  ) ? The tables needs to be normalize,   for exampe table 1.   SD or s.d.? better to make it consistent.
  • The Pre-post comparison is based on the  assumption that the subjects' social-demographic status remain unchanged. whether it could be demonstrated in this paper.
  • In the discussion part, more explanation on the improvement in maternal mental health will be more useful in addition to the current statement as it is the key results of this paper.

Author Response

Cover letter explaining revisions

We thank Reviewer 2 for their helpful comments and suggestions.  Responses to Comments and Suggestions for Authors are below, with numbering added for convenience.

Derek Bolton, on behalf of all co-authors

We note the opening comments: “Overall the paper is well written, with project settings, interventions, measures and results clearly presented. However, there are still a few areas to be improved.”

COMMENT/SUGGESTION #1: In the introduction part, more evidence on the role of health education and social support in improving maternal health is recommended, as it provides a theoretical basis for this project design, as well as a review of the existing evidence on this paper hypothesis. 

AUTHORS’ RESPONSE: Additional references have been added, lines 40-43. We have not included reviews of health education and social support as these topics and literatures are each very large, and out of scope of this report, which, as we say in the Introduction, is to report an evaluation of a new service, lines 69-79. 

COMMENT/SUGGESTION #1: Data analysis description not enough, please more detail, for example,  if t-test is a good chioce when mean more than SD ( table 4 "Low literacy sub-group"  ) ? The tables needs to be normalize,   for exampe table 1.   SD or s.d.? better to make it consistent.

AUTHORS’ RESPONSE: The main assumption of t-tests is that the data are approximately normally distributed, and this does not require that the mean be less than the standard deviation (the standardized normal distribution has mean zero and SD 1). We have corrected the inconsistency in the abbreviation of “standard deviation” in Table 4.

COMMENT/SUGGESTION #1: The Pre-post comparison is based on the  assumption that the subjects' social-demographic status remain unchanged. whether it could be demonstrated in this paper.

AUTHORS’ RESPONSE: it is usual in evaluating health interventions to assess social demographic status at baseline only in order to characterize the sample (e.g. to enable generalization or comparison with other studies). Sociodemographic status would only be assessed again post-intervention if it was an outcome variable and the intervention was designed to change it.

COMMENT/SUGGESTION #1: In the discussion part, more explanation on the improvement in maternal mental health will be more useful in addition to the current statement as it is the key results of this paper.

AUTHORS’ RESPONSE: Thank you for this suggestion. We have added more discussion of this, lines 520-22. The study was not designed to assess specific active ingredients, but we note that these results suggest that participation in the project is effective in improving the maternal mental health (lines 522-23).

Round 2

Reviewer 2 Report

After reading your quickly and exactly  response, I found the comments had being accepted  rightly.

There is no more suggestion, hope every thing smoothly!

This manuscript is a resubmission of an earlier submission. The following is a list of the peer review reports and author responses from that submission.

Round 1

Reviewer 1 Report

Dear Authors,

the article is very interesting and I was pleased to read it. There is one major issue and little notes to improve your article.

Major issue: Neither in the introduction nor in the discussion the project is placed in the current debate. This means that there are no indications whether there are already projects in this direction and where the research gap exists, or where the benefit of your project will be, which has not yet been verified. This should be discussed more clearly both in the introduction and in point 4. 

Little notes:

Please mention, what 'wte' stands for (page 3).

Do the social support groups ran weekly for 2 hours during the whole time (I think so, but it is not clear) (Page 3)

The mention of the power calculation is very good. However, a scale (GHLQ-12) is mentioned which does not appear in the whole text. Why was it used as a benchmark?

Why do you not use a control group? Why did you not decide on a third date (immediately after the intervention)?

Page 4: You say, engage (or reach), definied in various ways - Do you have one or two examples?

Does it makes sensce to put Point 2.6 in Point 2.4? There are information that are very useful to know before the measurement description.

Results: There are 58 re-assessed after 6-months. Why is engagement and the points after described for 61 mothers? The 3 drop-out does not interest.

Tables beginning with No.2: Please explain why it make sense for these sub-groups.

Health Literacy: Why there are only 55 mothers (not 58)?

I would be glad, if you edit your article, because the topic is very interested.

Reviewer 2 Report

Mechanics

Abstract: There are many imprecise statements and grammar errors in the abstract.

Line 19: The definite article missing before population,….it should read “the” population

Line 19th. The whole sentence require grammar revision. It should read. “Health improvement at the population level requires optimizing collaborations between statutory health services, civic organizations (e.g. churches, schools), and community groups in partnership with parents. Take off “themselves” that is a redundant term.

Line 21: “some” key elements. The word in front of “key” that identifies the key elements is missing. You could also say “Two” key elements……since you named two

Line 27. The word “and” is missing between mental health measures “and” health literacy

Line 27 the end of the sentence is incomplete and poorly formed

Line 29: Instead of saying, “in demographic terms” be more descriptive….state what the demographic is…this is too vague of a phrase to use for an abstract that must provide details

Lines 31 and 32: increase accessibility and improvement in mothers’ mental health outcomes are two separate concepts. State them separately

Introduction:

Line 56:  Take out the word “themselves” after mothers. If you are referring to mothers it is already themselves.

Line 66: Take out the word “firstly” and “secondly”. Instead you can state there were two main objectives of the PACT project: 1). To use community…….. and 2). To improve maternal etc.

Lines 70-73 is a run-on sentence.

These mechanical errors occur throughout. Have someone proofread for sentence structure and phrasing.

Content and Organization

 Introduction:

The last paragraph in the introduction inappropriately reports the results and the conclusions drawn base don the. Take these statements out.. Results should be reported in the results section and conclusions under conclusions/implications.

There needs to be more discussion about the existing community engagement programs and projects and how this one fills a gap in the literature. This literature review does not identify the gaps that this particular community engagement project no fills.

Materials and Methods:

This section needs development. Name the approaches that were used? CBPR? PAR?

You need a section on the sample…participants/subjects. Instead, you have  dispersed a discussion of the subjects throughout the section on evaluation and setting and engagement. You are really describing the under evaluation. Make a separate sample section

Under Evaluation 2.4 section lines 163 to 167 this information is part of the procedures. Please place this information in procedures

Measures:

Much of the information under sociodemographic data references the aims. That should be in the section on aims. Just talk about how you are measuring demographics in this section and not why you chose that population. This is the measurement section. That other information should go under a section about the sample.

Reliability

You need to report the reliability that you received from each of these measures. Did you compute an alpha coefficient??

Validity:

There is no information about the validity of the measures used except that you are just saying they are widely used. Any evidence of their validity and reliability with your particular population?

You need a data analysis section for the plan on how you are analyzing the quantitative data. What tests, how to measure reliability validity, any tests for assumptions skewness, kurtosis etc.

Results:

The format for the demographic tables needs work. What is the number of participants who responded to each category? Was there any missing data?

Overall very important work. Needs some grammatical work and some content is missing to make the report comprehensive